# Housing First and Single-Site Housing

**Patricia M. Chen**

Department of Public Policy and Political Economy, The University of Texas at Dallas, 800 West Campbell Road, GR 31, Richardson, TX 75080-3021, USA; pmpan@utdallas.edu

**Abstract:** In 2002, the United States embraced the Housing First approach, which led to the widespread adoption of this approach in cities across the nation. This resulted in programmatic variations of Housing First and calls for clarity about the Housing First model. This study uses a comparative case study approach to explore the differences across Housing First programs in five selected cities: Dallas, Austin, Houston, Los Angeles, and Salt Lake City. It focuses on one aspect of programmatic variation: housing type. Data collection consisted of in-depth interviews with 53 participants, documentation review, and site visits. Findings show differences in the type of housing used and explore the reasons why Housing First programs select such housing configurations. The results highlight how programmatic variation does not necessarily mean the Housing First model lacks clarity. Rather, homeless service providers adapt the model to address local challenges and needs, resulting in the variation seen across programs and cities. The findings elucidate the debate about variation in the Housing First model and the call for fidelity.

**Keywords:** chronic homelessness; Housing First; homelessness policy

## 1. Introduction

In 2002, the United States shifted its approach to homelessness to Housing First. This was a significant departure from the previous traditional, linear continuum of care model, known as Treatment First, which required homeless individuals to complete a sequence of steps to demonstrate their "housing readiness". Housing First is an approach where homeless individuals are placed into permanent housing as quickly as possible. There is a separation of housing from services, meaning housing placement and tenure is not conditional on prerequisites such as sobriety or medication adherence (Tsemberis and Asmussen 1999). While support services are offered to homeless individuals (or "clients"), clients themselves decide whether to participate and the extent of that participation. The underlying premise of the Housing First approach is that homeless individuals need the safety and stability of permanent housing to address challenges, such as mental health problems or substance use disorder.

In 2010, the Obama administration adopted Opening Doors, the first comprehensive, strategic federal initiative to prevent and end homelessness; among its objectives was the goal to end chronic homelessness by 2015. Opening Doors laid out the Obama administration's support for Housing First, with the statement, "For people experiencing chronic homelessness, the research is clear that permanent supportive housing using a Housing First approach is the solution" (United States Interagency Council on Homelessness 2010). The U.S. Interagency Council on Homelessness (USICH) embracing the Housing First approach resulted in widespread adoption of this approach in cities across the nation.

However, this widespread adoption of Housing First programs across the nation has resulted in programmatic variations and calls for clarity about the Housing First approach (Rog et al. 2014; Tabol et al. 2010; Pleace 2011). This study uses a comparative case study approach to explore the differences across Housing First programs in five selected cities. The programs varied in several aspects

such as housing type and service provision. This paper focuses on one aspect of programmatic variation: housing type. The data and findings presented here are a subset of a larger project, which identified the differences across Housing First programs and contextual factors affecting the implementation of Housing First and its subsequent variation. I begin with an overview of the literature followed by the study's methodology and discussion of the results. The findings elucidate the debate about variation in the Housing First model and the call for fidelity.

## 2. Literature

The origin of the Housing First model can be traced back to Sam Tsemberis. Tsemberis developed the Consumer Preference Supported Housing model, also referred to as the Pathways Housing First model. The Pathways Housing First model is based on several principles: (a) that homeless individuals with mental illness maintain independent housing with appropriate supports; (b) the homeless should be treated as consumers, meaning they ought to have choice; (c) housing is not conditional on sobriety, medication adherence, or any form of treatment; (d) the use of scattered-site housing; (e) services are offered by Assertive Community Treatment (ACT) teams in the community; (f) the use of a harm reduction approach for substance use disorder; and (g) the type, frequency, and sequence of treatment is determined by the individual (Tsemberis and Asmussen 1999; Tsemberis et al. 2004). The Pathways Housing First program originated in New York, but the Housing First model was quickly adopted by other cities. Pathways to Housing expanded to Philadelphia, Washington, D.C., and Calgary, Canada. Utah became the first state to adopt the Pathways Housing First model statewide (McEvers 2015). Pathways has also worked with the Department of Veterans Affairs to expand the Housing First model to help homeless veterans (Montgomery et al. 2014). The Substance Abuse and Mental Health Services Administration considers Housing First an evidence-based practice.

Much of the literature on Housing First has focused on outcomes for homeless clients, such as reduced utilization of shelter, healthcare, and criminal justice systems and increased housing stability (Collins et al. 2013; Culhane et al. 2002; Gilmer et al. 2010; Mares and Rosenheck 2010; Stefancic and Tsemberis 2007; Stergiopoulos et al. 2015). Findings from these studies may compel policymakers to support Housing First by presenting it as a cost-effective solution to chronic homelessness. While studies have shown the positive impact of Housing First for the chronically homeless, the variation across programs has raised concerns about the need for clarity and resulted in subsequent fidelity assessments.

The calls for clarity are not without merit. (Padgett et al. 2016) noted that the rapid growth of Housing First programs resulted in different versions of Housing First. In their review of the literature on supported housing, (Tabol et al. 2010) conclude that there is a need for greater model clarity and a fidelity instrument that would allow for comparisons across programs (Tabol et al. 2010). Similarly, (Rog et al. 2014) review found that the evidence for permanent supportive housing is promising, but "there is a need to determine fidelity to permanent supportive housing principles, operationalize interventions, and examine program components that are related to outcomes". (Rog et al. 2014; Stefancic et al. 2013) acknowledged this variation across Housing First programs, noting "the effectiveness and advocacy surrounding Housing First has spurred both the creation of programs similar to that developed by Pathways, as well as programs that exhibit considerable variation from the original model" (Stefancic et al. 2013). Pleace (2011) also calls for the need for a better understanding of the variation in Housing First services, given the "model drift" from the Pathways to Housing First approach (Pleace 2011). Tsai and Rosenheck (2012) echo Pleace and urge caution about the widespread adoption of Housing First without exploring other models (Tsai and Rosenheck 2012).

With variation across programs and the demand for clarity, there are an increasing number of studies on the implementation of Housing First programs, including fidelity assessments (Aubry et al. 2015; Gilmer et al. 2013; Goering et al. 2016; Macnaughton et al. 2015; Nelson et al. 2014). For example, Stefancic et al. (2013) developed the Pathways Housing First Fidelity Scale to assess the degree to which programs adhere to the Pathways Housing First model. The fidelity measure has 38 items across five domains: housing choice and structure, separation of housing and treatment, service philosophy,

service array, and program structure (Padgett et al. 2016; Stefancic et al. 2013). USICH also developed a Housing First checklist, which allows service providers to assess their programmatic alignment with Housing First principles. This checklist includes eleven items and, much like the Pathways Housing First model, USICH emphasizes low barrier to housing entry, the separation of housing and treatment, consumer choice, and the use of a harm reduction approach (United States Interagency Council on Homelessness 2016).

Others examined the variation by categorizing Housing First programs based on the descriptions of services offered. Pleace and Bretherton (2012) created a broad taxonomy of Housing First services with three categories: "pure" Pathways Housing First services, Communal Housing First Services, and Housing First "Light" services (Pleace and Bretherton 2012). While variation exists, they note that the latter two groups are influenced by or closely aligned with the Pathways paradigm and that the term "Housing First" can refer to a specific type of service, such as the Pathways model, as well as a broader concept.

The fidelity assessments and categorization of Housing First services highlight the variation across programs and underscore the need for clarity on the Housing First approach. However, much of the literature focuses on comparisons with the Pathways Housing First model and few studies consider the differing aspects of Housing First programs, such as the type and location of housing, and whether it affects outcomes for program participants. Many of the studies that consider housing features and the type of placement primarily focus on psychiatric patients and not the chronically homeless (Linn et al. 1980; Nelson et al. 1998). To date, few studies have examined the type of housing placement (Harkness et al. 2004; Somers et al. 2017; Whittaker et al. 2016, 2017). Furthermore, there is little research that examines why such variation across Housing First programs occurs.

## 3. Methodology

This study used a comparative case study approach to explore the variation across Housing First programs in five selected cities: Dallas, Texas; Austin, Texas; Houston, Texas; Salt Lake City, Utah; and Los Angeles, California. Austin and Houston were selected for in-state comparisons with Dallas; Los Angeles for the magnitude of its homeless population; and Salt Lake City because it is often hailed as an example of the success of the Housing First model. The nature of the research question necessitated a qualitative approach. The comparative case study method was exploratory, considered the case in its real-life, contemporary context, and utilized in-depth data collection (Creswell 2013). The methodology allowed for detailed findings and the discovery of nuance and complexity for each case.

Data were collected from in-depth interviews with participants, site visits, and documentation review. Fifty-three participants were interviewed. Participants included CEOs, program administrators, case managers, and street outreach team members. The participants represent thirty-seven organizations and programs across the five cities. Participants' names and affiliations are not disclosed for confidentiality purposes. Interviews lasted approximately 45 to 60 min. Prospective participants were identified through an extensive online search for service providers in each city, relying on information from the lead agency for each Continuum of Care (CoC), news articles in local papers, and grant applications available on HUD Exchange. Participants were recruited via emails and phone calls. Snowball sampling was used to identify other major service providers and key individuals within each CoC. This technique also served as an indicator of data saturation because similar names would be suggested for prospective participants. Additionally, this allowed me to verify that major service providers and key individuals in each CoC had been identified and offered the opportunity to participate in the study. Interviews were transcribed and coded using NVivo. The review and analysis of interview transcripts generated codes and themes identified within each city as well as cross-case themes. Information from participant interviews, such as dates of certain events (e.g., introduction of a Coordinated Assessment System, also known as Coordinated Access or Coordinated Entry),

were cross-referenced with available documentation, such as annual reports, CoC meeting minutes, news articles, press releases and the like for validation purposes.

I also conducted site visits to as many programs as possible across the five cities. These visits typically entailed meeting with homeless service providers, observing meetings and providers' interactions with staff, other organizations, and homeless clients, and touring housing sites and the surrounding neighborhoods. Site visits to Houston, Los Angeles, and Salt Lake City were conducted in July 2016; Austin in September 2015; and Dallas in November/December 2015 and again in August 2016. During the study timeframe, the Dallas Commission on Homelessness held a series of public meetings to discuss the city's homeless problem, obtain community feedback, and develop a strategy. I attended these meetings and observed the interaction among Commissioners and community responses to proposed ideas.

## 4. Housing Type: Scattered-Site versus Single-Site

The adoption of single-site housing models has raised questions about fidelity to the Housing First approach and whether different housing configurations matter. The Pathways Housing First model uses scattered-site housing because it sidesteps potential NIMBY (Not In My Back Yard) issues and promotes community integration (Tsemberis and Asmussen 1999). Programs in this study used scattered and single-site housing, or a combination of the two, though certain types of housing were more prevalent in certain cities. Programs in Texas primarily used scattered-site housing, particularly in Dallas. Some programs in Austin and Houston used on single-site housing, but these were exceptions. Texas participants perceived NIMBY as an influential factor in their program's housing configuration. A participant in Austin candidly noted, "I think the housing options [single- or scattered-site] are based on what is palatable to the community".

Participants noted the difficulty of developing single-site housing because of NIMBY. One program spent years trying to find suitable location for a housing development, but ultimately relocated outside city limits because of intense community opposition. Another program had an existing substance abuse program in a local community and decided to expand its operations by developing a housing property for its homeless clients. While the community was generally supportive of the treatment program, they raised concerns about the prospect of having homeless neighbors. This participant astutely noted this tension, "It is one thing to have [homeless] people on-site for treatment once a week versus living there". Her program met with neighborhood associations with the hopes of convincing the community that the housing program and its homeless residents would be good neighbors.

A recurring theme among participants in Texas was the notion of stigma and a desire to avoid concentrating homeless residents in a single building or community. Many participants discussed the need for housing and residents to blend in with the neighborhood. As one participant noted, "Scattered-site housing is better. On the political side, it is easier for scattered-site housing to blend in with the rest of the neighborhood". Another participant described how single-site housing could be appropriate for certain subpopulations such as those with chronic substance use disorder, but it would be better to avoid the stigma of the single-site model.

Texas participants shared how their programs sought to make the housing property and its homeless residents integrated into their respective communities. For example, one program in Houston convened a neighborhood panel and asked for input on the design of the building. This participant described how the program selected a specific roof color instead of terracotta tiles, so the aesthetics of the building would match other properties in the community. A participant in Austin noted the importance of integrating homeless residents into the community, "We do want to increase interaction with the non-homeless and the surrounding community. We plan on having gardening projects and other ways for our residents to blend into the community. We want folks to become integrated into the community". Such integration may dispel misconceptions of homeless individuals and increase the community's acceptance of the housing property and its residents. Both these statements reveal how participants viewed the necessity of integration of the property and homeless residents to overcome

community resistance. Participants also felt compelled to demonstrate their program's value and justify their presence in the community. While NIMBY was a recurring problem raised by participants across all five cities, it appeared most problematic for Dallas. A participant in Dallas shared, "People support Housing First and permanent supportive housing programs as long as it is in downtown and not in their neighborhood". Many homeless service providers, including the city's shelter, are located in the downtown area. Participants in Dallas cited fears of community opposition in the form of BANANA (Build Absolutely Nothing Anywhere Near Anything), a more extreme version of NIMBY. One example of BANANA is the Dignity Field proposal, which was authored by a local resident in response to Dallas' growing homeless population. This proposal recommended relocating the homeless population away from downtown and the surrounding area. Specifically, it proposed re-purposing an old naval station into a large homeless encampment, called Dignity Field, using a "Care First" (i.e., Treatment First) approach. Opponents referred to Dignity Field as a homeless concentration camp and raised concerns about access to services for the homeless, given the camp's remote location. Although the City Council ultimately rejected the Dignity Field proposal, this example highlights the extent of BANANA in Dallas and community resistance perceived by participants.

The severity of NIMBY in Dallas was even noted by fellow Texas participants though they face this problem themselves, albeit to a lesser extent. As one participant in Houston noted, "NIMBY is not as big of a problem in Houston as it is in Dallas, but it is still one of the biggest problems". This participant also noted how Houston benefitted from the lack of zoning ordinances, which allowed housing to be located in various neighborhoods:

> Houston has different zoning and so people are used to different buildings going up in the neighborhood. Over here, there's a preschool and a bar. They share a wall and are in the same building, so I think people are used to this.

While Texas programs wrestled with NIMBY and preferred scattered-site housing, programs in Los Angeles and Salt Lake City tended to use any housing available. Across the five cities, single-site housing was the most prevalent in Los Angeles. Service providers in Los Angeles embraced the single-site model due to the dire need for housing because of its large homeless population. The magnitude of homelessness was a recurring theme among participants in Los Angeles. Participants shared a sense of urgency and frustration about the growing homeless population, especially those service providers working in Skid Row. According to the annual Point In Time count, the total homeless population in the city and county of Los Angeles increased from 43,854 to 55,188 individuals between 2016 and 2017 (United States Department of Housing and Urban Development 2017). In addition to a large homeless population, participants also shared concerns about the shortage of affordable housing and low vacancy rates as reasons for utilizing any housing, regardless of its configuration. In response to questions about housing type, one participant remarked, "We don't have the luxury with the housing shortage to be ideological". Participants also recognized that Los Angeles would not be able to build its way out of the homeless problem, which is why some programs disregarded specifications about housing type and used whatever housing is available for clients. Overall, participants in Los Angeles emphasized the need to move homeless clients into housing regardless of the type of housing.

This sentiment was echoed by participants in Salt Lake City. The different housing configurations in Salt Lake City are particularly interesting because Utah was the first state to adopt the Pathways Housing First model statewide. As such, one would expect high fidelity to the Pathways Housing First model. Service providers adhered to the Pathways model, emphasizing principles such as client choice. However, some programs differed from the Pathways model on aspects such as housing type and treatment approach (e.g., Intensive Case Management rather than Assertive Community Treatment). Participants also described the Housing First implementation process as centrally-led and locally-developed, which allowed programs to adapt Housing First to its local context. Findings indicate two factors contributed to the different types of housing in Salt Lake City: a desire to house the homeless as quickly as possible and a centrally-led, locally developed implementation process for Housing First. A participant from Salt Lake City noted homeless clients could be housed faster with



scattered-site models that rely on existing housing stock rather than waiting on the completion of a single-site housing development. This participant also shared:

> We are developing some congregate-site for affordable housing. There are designated units within an apartment complex, but only about 10 units here or there. We aren't getting the magnitude we need.

This statement shows how this program sought to move homeless clients into housing as quickly as possible regardless of the housing configuration. While this participant acknowledged the benefits of scattered-site housing, he recognized there were not enough available scattered-site units to meet the demands of the homeless population; this program used scattered- and single-site housing. Participants offered a variety of reasons for the type of housing used for their respective programs. Texas participants perceived NIMBY as a significant challenge and primarily used scattered-site housing to avoid NIMBY issues. In Los Angeles, programs were driven by the shortage of affordable housing and the magnitude of the homeless population and resorted to using any housing available. Programs in Salt Lake City had a similar approach, emphasizing the rapid housing placement over the housing configuration unless addressing a specific client need or request.

*Single-Site Housing and Fidelity to Housing First*

The discussion in this section focuses on the programs that used single-site housing, exploring their perceptions about fidelity to Housing First. Participants listed several advantages of single-site housing: increased programmatic control, fostering a sense of community, and ease of access to services for homeless clients (e.g., on-site case managers and support services). Furthermore, they believed single-site housing allowed for greater fidelity to Housing First principles.

Single-site housing programs using a blended management model had the most programmatic control compared to other programs. These programs were prevalent in Los Angeles and Salt Lake City; Houston had one program that operated this way. Blended management refers to the organizational model that combines service provision and property management. Both functions are within the same organization and are intended to work together but are kept separate through organizational structure, such as having different chains of commands. These programs also typically developed and/or owned the property they managed. Because these programs do not rely on partnerships with private landlords to house clients, they effectively eliminated barriers to housing entry for their homeless clients. One participant shared how her program was more lenient with the background check process for homeless clients because of their barriers to housing entry (e.g., poor rental history, bad credit or existing debt). Participants also noted that their blended management models had an explicit expectation for property management and support services to work together to ensure that the client retained housing. This is a stark contrast to scattered-site programs who were at the mercy of property management. As one participant of a scattered-site housing program explained:

> We don't run the building. Another group runs the housing site and their rule is that any alcohol or drug use is against the house rules. So, we explain that to clients that is the property management's rule not ours and that they might get evicted.

In this situation, the homeless service provider embraced Housing First principles, but the property management's rules made it a de facto Treatment First program. Property management erected rules and barriers beyond the service provider's control, which conflicted with Housing First principles. This highlights how the scattered-site model and its reliance on partnerships with private landlords can affect the program's fidelity to Housing First. It also shows how single-site housing programs using a blended management model can have greater programmatic control and fidelity to Housing First. For programs using single-site housing, participants also emphasized the importance of creating community. They believed single-site housing would address client concerns about loneliness and social isolation. Some programs, such as those in Skid Row, made strategic decisions to house clients

close to their friends (i.e., same or nearby property). Service providers recognized the value of these relationships to their clients and sought to preserve those social networks. One participant shared how his program carefully crafted community:

> We work really hard to create community in Skid Row . . . People are connected with service providers—it's all here—and their friends are here, so moving away from Skid Row isn't something they want to do because their support system is all in this area.

Similarly, a participant in Austin noticed that her program's homeless clients struggled in scattered-site housing. They complained about being socially isolated and received numerous lease violations for having unauthorized guests. As she explained, "They [the homeless clients] just can't say no to their friends from the street and [there is] the lack of community . . . With [property name] we will have controlled access in terms of visitors and can create a sense of community". At the time of the interview, this program had just begun developing a single-site housing project. This participant's statement reveals how homeless clients desired community and her program sought to provide it by using single-site housing.

The importance of community and ease of access to services were recurring themes among single-site housing programs. Some programs chose to have on-site healthcare providers, while others had support services located nearby. Two programs, one in Salt Lake City and another in Los Angeles, had health clinics within walking distance of their housing. While participants strongly adhered to the Housing First principle of the separation of housing and service participation, they believed the proximity to services would remove barriers to access.

Some participants felt compelled to justify their program's use of single-site housing, as if it contradicted the Housing First approach. One participant shared how skeptics questioned if his single-site housing program could still be considered Housing First. The underlying implication of such criticism is that single-site housing and the Housing First approach are mutually exclusive; findings from this study show they are not. This participant explained how his program integrated Housing First principles into single-site housing model, stating:

> It is the physical manifestation of Housing First by *design*. It's not just accepting the concept, but how it permeates. If you really listen to the [formerly homeless] residents, it can really inform the design.

This participant described how residents requested a community garden and more open spaces, which his program subsequently provided. He believed that this gave residents a sense of ownership, cultivated community, and expanded on the Housing First principles of consumer choice and respect. This example demonstrates how Housing First principles can be applied in a congregate setting. These findings echo the results of the study of a single-site Housing First program by (Stahl et al. 2016). Stahl et al. (2016) used a grounded theory approach to study the experience of formerly homeless individuals in a single-site Housing First program. Their findings suggest that the sense of community and stability of single-site Housing First programs may also support housing retention and stability (Stahl et al. 2016). While single-site housing deviates from the Pathways model, the findings here show that programs using different types of housing can still adhere to the Housing First approach.

## 5. Limitations

This study has several limitations. The findings may not be generalizable to Housing First programs across the nation, given the small sample size and purposive selection of cities for study. However, the results may be insightful to cities with comparable contextual features and/or Housing First programs facing similar challenges. Additionally, the findings are based on the perspective of homeless service providers. Future research would benefit from incorporating the views of property management and homeless individuals.

## 6. Discussion

Across the five cities, there was variation in the implementation of Housing First. This paper focused on the differences in housing type and explored reasons that influenced the program's selection of such housing configurations. Texas participants perceived NIMBY as a significant obstacle and preferred scattered-site housing, whereas Los Angeles and Salt Lake City utilized any housing available, regardless of its configuration. The variation across Housing First programs does not mean the model lacks clarity. Rather, homeless service providers are adapting Housing First to address local challenges and needs. Participants in this study emphasized the importance of Housing First principles such as consumer choice and the separation of housing and treatment even though their programs differed in several aspects.

Critics have questioned whether programs using different housing configurations can maintain fidelity to Housing First. The type of housing itself does not appear to matter as long as programs adhere to Housing First principles. Single-site housing had several advantages: greater programmatic control, ease of access to services, and fostering community. The findings show single-site housing and Housing First are not mutually exclusive. If the Housing First approach is client-centered and client-driven, there should be an array of housing options dependent on the needs and preferences of homeless clients. The Housing First discussion would benefit from examining other programmatic differences. Future research should consider the location of housing and examine whether neighborhood features affect the outcomes of homeless clients.

**Funding:** This research was funded by the Office of Graduate Studies at the University of Texas at Dallas.

**Acknowledgments:** The author would like to thank the research participants for their time and contributions to the study.

**Conflicts of Interest:** The author declares no conflict of interest.

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
