# Peer review of "Housing First and Single-Site Housing"

_socsci, doi:10.3390/socsci8040129_

Round 1

Reviewer 1 Report

Overall, this is a well written manuscript. There are a couple of changes that could enhance the article, one of which is important and the other not as essential.

The literature review could benefit from an expanded discussion of the "pathways" approach. You do mention more details later in the paper, but it is central issue that requires attention so the reader can follow your paper. (required)

The idea that housing first has been promoted as a panacea could be elucidated somewhat more than a single sentence. (optional)

The analysis could benefit from further identification of emergent themes. Based on your method section, you should have a significant amount of data that could be mined for themes related to effective use of housing, NIMBY aversion or other issues related to placement and community resistance/acceptance. The list could be quite extensive. (strongly recommended)

Author Response

Thank you for these constructive comments. 

Re: discussion about the Pathways approach

Thank you for pointing this out.  I concur and added a discussion about the Pathways Housing First model to the literature review. 

Re: Discussion of other emergent themes

I expanded the discussion on single-site versus scattered-site housing. This includes more on the topic of NIMBY. It also discusses how homeless service providers wanted their housing and homeless clients to blend in with the community.  The other themes that emerged from the qualitative data relate to challenges and responses of service providers in service delivery.  I excluded discussing these themes because it does not fit with the focus of this paper. 

Reviewer 2 Report

The paper explores differences across Housing First programs in five US cities and discusses differences in the type of housing used and the reasons behind housing type choices. It claims that HF model is adapted to suit local circumstances and this variation doesn’t necessarily equate to a lack of clarity regarding the HF model.

The paper is well presented and focused on fulfilling its stated aims. Apart from some very minor requests listed below, my only substantial comment relates to the depth of the analysis. The paper does, indeed, discuss differences in housing type and the rationale behind such differences, but the discussion itself is relatively short. Given the substantial qualitative material (approx. 50 interviews with key actors), I would have thought the author(s) could provide a richer account of key informant perspectives. I don’t doubt the conclusions that the author arrives at, but the depth/nuance of insight into decision-making processes surrounding housing type is lacking. Some further illustrative detail (subject to word limit restrictions) would enliven the paper, and buttress the arguments made by the author(s).

P1 l7: which ‘federal government’? (same comment applies to other instances in the paper)

P1 l40: ‘critics claiming the Housing First model lacks clarity’. Please add citations so that the reader can find the accounts of said critics

P2 l 89: ‘Others examined by the variation by categorizing’. Rephrase for clarity

Author Response

Thank you for this helpful feedback.  I appreciate these constructive comments and the opportunity to respond. 

Re: Depth of the analysis

I expanded the section on single-site versus scattered-site housing, which includes a bit more on the topic of NIMBY.  I also discuss how homeless service providers wanted their housing and homeless clients to be integrated into the community where the housing is located.  The qualitative data presented here are a subset of the findings from a larger project, which focused on identifying the variation across Housing First programs and examining the contextual factors affecting the implementation of Housing First.  I make note of this in the introduction, but if needed (or recommended), I can also add this to the data and methodology section.  In my data collection efforts for the larger project, I identified that housing configuration was one aspect in which Housing First programs varied within and across my five cities.  The other themes that emerged from the qualitative data (from the larger project) relate to the challenges faced by providers in service delivery and their respective responses.  I excluded those themes because it does not fit with the focus of this paper.    

 Re: which federal government

I revised the references to “federal government” to indicate the United States or a specific agency of the United States government (e.g., U.S. Interagency Council on Homelessness).

Re: critics claim the Housing First model lacks clarity

The critics I initially referenced were individuals I encountered during my fieldwork and claimed the Housing First model lacked clarity.  I rephrased this sentence to better reflect my intent, by removing the word critic and adding "calls for clarity and subsequent fidelity assessments." The citations for the calls for clarity and fidelity assessments are following paragraphs in the literature review section. 

Re: “Others examined by the variation by categorizing”

Thank you for catching this. This sentence was poorly worded and I have rephrased it for clarity. 

Reviewer 3 Report

This short paper adds some concrete examples on the meaning of fidelity in Housing First programmes, which is a useful addition to the literature.  

Author Response

Thank you for this feedback. 

Round 2

Reviewer 2 Report

I believe the author(s) has addressed my previous concerns and I'm happy to recommend the paper be considered for publication